# Relation between Heart Rate Variability and Disease Course in Multiple Sclerosis

**DOI:** 10.3390/jcm9010003

**Published:** 2019-12-18

**Authors:** Tatjana Reynders, Yori Gidron, Jella De Ville, Maria Bjerke, Ilse Weets, Ann Van Remoortel, Lindsay Devolder, Miguel D’haeseleer, Jacques De Keyser, Guy Nagels, Marie B. D’hooghe

**Affiliations:** 1Department of Neurology, Universitair Ziekenhuis Antwerpen, Wilrijkstraat 10, 2650 Edegem (Antwerp), Belgium; tatjana_reynders@live.be; 2Department of Neurology, Universitair Ziekenhuis Brussel, Laarbeeklaan 101, 1090 Jette (Brussels), Belgium; jella.deville@gmail.com (J.D.V.); miguel.dhaeseleer@mscenter.be (M.D.); jacquesdekeyser@gmail.com (J.D.K.); guy.nagels@gmail.com (G.N.); 3Center for Neurosciences, Vrije Universiteit Brussel, Laarbeeklaan 103, 1090 Jette (Brussels), Belgium; ygidron@univ.haifa.ac.il (Y.G.); maria.bjerke@uzbrussel.be (M.B.); devolderlindsay@gmail.com (L.D.); 4School of Nursing, University of Haifa, Abba Hushi Boulevard 199, 3498838 Haifa, Israel; 5Laboratory of Neurochemistry, Universitair Ziekenhuis Brussel, Laarbeeklaan 101, 1090 Jette (Brussels), Belgium; ilse.weets@uzbrussel.be; 6BIODEM, Institute Born-Bunge, Universiteit Antwerpen, Universiteitsplein 1, 2610 Wilrijk (Antwerp), Belgium; 7National Multiple Sclerosis Center, Vanheylenstraat 16, 1820 Melsbroek (Vlaams-Brabant), Belgium; ann.vanremoortel@mscenter.be; 8Rega Institute, Department of Microbiology and Immunology, Katholieke Universiteit Leuven, Oude Markt 13, 3000 Leuven (Vlaams-Brabant), Belgium; 9VIB, Center of Microbiology, Oude Markt 13, 3000 Leuven (Vlaams-Brabant), Belgium; 10Department of Neurology, Universitair Medisch Centrum Groningen, Hanzeplein 1, 9713 GZ Groningen (Groningen), The Netherlands; 11Faculté de Psychologie et des Sciences de l’Education, Université de Mons-Hainaut, Place du Parc 20, 7000 Mons (Hainaut), Belgium

**Keywords:** multiple sclerosis, autonomic dysfunction, heart rate variability, relapse

## Abstract

Little is known about the interplay between the autonomic nervous system and disease activity in multiple sclerosis (MS). We examined the relationship between heart rate variability (HRV), a reliable measure of vagal nerve function, and disease characteristics in a prospective MS cohort. Standard deviation of each normal-to-normal inter-beat interval (SDNN) and root mean square of successive differences (RMSSD), global indices of HRV, were measured in 114 MS patients, which included four predefined subgroups, and 30 age and sex-matched healthy controls (HC). We assessed group differences at baseline, HRV reproducibility at month 3, and used logistic regression modeling to relate baseline HRV with relapse occurrence. No significant HRV differences were found between MS and HC and between MS subgroups. In MS patients, both HRV indices correlated with age (r = −0.278, *p* = 0.018 and r = −0.319, *p* < 0.001, respectively) and with month 3 assessments (r = 0.695 and r = 0.760, *p* < 0.001). Higher SDNN and RMSSD at baseline were associated with self-reported relapses at month 3 (OR = 1.053, 95% CI (1.013–1.095), *p* = 0.009 and OR = 1.065, 95% CI (1.016–1.117), *p* = 0.009), and SDNN at baseline with relapses at month 12 (OR = 1.034, 95% CI (1.009–1.059), *p* = 0.008; ROC, AUC = 0.733, *p* = 0.002). There were no baseline HRV differences between MS and HC or between subgroups. Post-hoc analysis showed an association with an increased relapse risk.

## 1. Introduction

Multiple sclerosis (MS) a complex, chronic disease of the central nervous system (CNS), is known for its wide clinical, radiological, and pathological heterogeneity including a varying degree of inflammation and neurodegeneration [1]. Most patients start with a relapsing-remitting course, which may be followed by a secondary progressive phase. A minority of subjects present with an insidious progression of disability from onset, or primary progressive [2].

Autonomic symptoms are prevalent in MS [3,4,5]. Based on the immunoregulatory functions of the autonomic nervous system (ANS) [6], ANS dysfunction might play a role in the pathogenesis and progression of MS. The vagal nerve, responsible for the main parasympathetic output, directly communicates with the brain and can reduce peripheral inflammation in experimental conditions [7]. While some beneficial effects of vagal stimulation have been demonstrated in distinct inflammatory diseases such as rheumatoid arthritis and inflammatory bowel disease [8,9], no data are available in MS. The relation between ANS dysfunction and disease activity/severity in MS remains controversial [10]. This is probably related to the complexity of interactions among multiple variables, lack of prospective studies, use of heterogeneous methods and the measured outcome parameters.

Heart rate variability (HRV), reflecting instantaneous parasympathetic modulation of cardiac conduction, was shown to be a potent and reliable tool to examine ANS function in vivo with strong correlations with vagal activity [11]. Several studies found lower HRV in MS patients than in healthy controls (HC) [10,12,13], while others found more pronounced (para)sympathetic dysfunction in progressive than in relapsing-remitting phenotypes [10,14,15]. Most studies did not account for the effect of age on HRV [16,17].

Improving our understanding of HRV modulation in relation to the clinical and treatment characteristics of MS, might be helpful for the identification of new prognostic biomarkers or for the development of treatment approaches in MS. We therefore investigated the potential of HRV as an indicator of heterogeneity with regard to the degree of inflammation, considering the vagal nerve’s modulatory role in inflammation. We set up a prospective cohort study in predefined subgroups of different MS phenotypes (including treated and untreated relapsing remitting (RRMS), primary progressive (PPMS), and benign course MS) and age- and sex-matched healthy controls (HC). We aimed to identify whether HRV measures are associated with MS (MS versus HC), with MS phenotype, treatment status, or measures of disability and severity. We focused on the standard deviation of each normal-to-normal inter-beat interval (SDNN) and the root mean square of successive differences (RMSSD), measures of parasympathetic function and global HRV [18,19], as most relevant HRV parameters [19,20], because they are simple to use and interpret, suitable for short-term recordings [19] and have shown a predictive value in other diseases [21,22]. In addition, SDNN increases with transcutaneous vagus nerve stimulation [20], proving that SDNN reflects parasympathetic activity [19].

High sensitive C-reactive protein (hsCRP) and neurofilament light (NFL) serum concentrations were assessed at baseline, as they respectively reflect peripheral inflammation and neuronal damage, respectively, and were previously used in MS studies [23,24]. In addition, hsCRP is related to SDNN in healthy volunteers [25]. We assessed the stability of SDNN and RMSSD over three months and investigated whether SDNN and RMSSD at baseline were associated with relapse occurrence during a follow-up of 3 and 12 months.

## 2. Materials and Methods

MS patients from the National MS Center Melsbroek (NMSC) and the University Hospital Brussel (UZB), diagnosed according to the 2010 revised McDonald criteria [26], were included. Onset phenotype, disease course modifiers [27], and treatment status delineated predefined subgroups (Table 1), including primary progressive (PP) and relapsing-remitting (RR) MS [28]. Based on the difficulty to separate inflammation from neurodegeneration in the secondary progressive phase of the disease, we did not include patients with secondary progressive MS in this study. We selected one class of injectable immunomodulatory drugs in order to obtain a homogenous treatment group. Hence, IFN-beta was chosen, taking into consideration that patients could be treated with different IFN preparations, resulting in a larger patient pool.

HC were recruited to match the age- and gender distributions of the whole MS cohort. The ethics committees of the UZB and the NMSC approved the study (B.U.N. 143201317985), conducted in compliance with the Declaration of Helsinki (October 2013), the Belgian law on experimental research in humans (5 July 2004) and the guidelines of Good Clinical Practice. All participants provided written informed consent.

This study employed a prospective design and included intergroup comparisons at baseline and assessment of self-reported relapses after three and twelve months. In line with previous studies [12,29], we aimed to include 30 patients per subgroup. Demographics, clinical variables (Expanded Disability Status Scale (EDSS) and the derived age-related MS severity score (ARMSS [30])), medication and comorbidity were assessed at each visit.

A non-invasive two-lead ECG registration (eMotion HRV, Motion Ltd., Kuopio, Finland) in supine position at rest, was performed at baseline (all MS and HC participants) and three months later (MS and HC recruited in the NMSC) for 10 min (1000 Hz sampling frequency). Kubios HRV 2.1 software (Department of Applied Physics, University of Eastern Finland, Kuopio, Finland) was used to select the middle 5 min interval (2′30″ to 7′30″), followed by ectopic interval detection (4th order filter) with cubic spline interpolation and de-trending with Wavelet Packet in HRVAS software [18] as in [31]. The batch processing function allowed a blinded analysis.

At baseline, each MS and HC participant provided a venous blood sample (10 mL EDTA tube), which was immediately centrifuged at 3500 rpm during 15 min at 3–5 °C. The supernatant was transferred to 1 mL cryotubes and stored in a −80 °C freezer. HsCRP and NFL concentrations were determined in a blinded fashion with high sensitivity assays. HsCRP concentrations were measured on a Vitros 4600 analyser (Ortho Clinical Diagnostics, Rochester, NY) using a immunoturbidimetric latex immunoassay in high sensitive mode (CRP Vario, distributed by Abbott, Wiesbaden, Germany) [Abott, Laboratories. C-reactive protein. 2008]. NFL concentrations were assessed on the Simoa^®^ HD-1 Analyzer (NF-LIGHT^®^, Quanterix Corporation, Boston, MA, USA) according to the manufacturer’s instructions [Quanterix, Corporation. Simoa NF-light advantage (SR-X) Kit. 2019]. The mean NFL intra-assay coefficient of variation (CV) of the patient samples was 4.7%, while the inter-assay CV of an internal quality control serum pool was 5.4%.

Variables with a non-Gaussian distribution were transformed to a normal distribution using the LG10-transformation and rechecked for normality using the Skewness and Kurtosis tests. Demographic and clinical intergroup differences at baseline were assessed with the independent samples t-test with Levene’s test for equality of variances (2 groups), one-way ANOVA (>2 groups, Tukey HSD post-hoc analysis) or Pearson chi-squared test (for categorical variables).

An independent samples t-test and one-way ANOVA compared SDNN and RMSSD intergroup differences at baseline between MS and HC, and between MS subgroups, respectively. Correlations between continuous and categorical variables in MS samples were performed using the Pearson correlation test and Pearson chi-squared test, respectively.

Reproducibility of HRV parameters at three months was assessed using the Pearson’s correlation and Wilcoxon signed ranks tests. A binary logistic regression model included SDNN or RMSSD at baseline as the independent variable, and relapse occurrence over three and twelve months (yes/no), respectively, as the dependent variable. After checking for collinearity (r ≥ 0.25 [32]), non-collinear variables (age, sex, and MS subgroup allocation) were forced into the multivariate model, while collinear variables EDSS and NFL were added as an expression of age. This was done by first performing a linear regression with EDSS or NFL as the dependent and age as the independent variable, then multiplying the collinear variables by their respective regression coefficient. A receiver operating characteristic (ROC) analysis was performed for SDNN at baseline and relapse occurrence over twelve months.

Statistical significance level was set at 0.05. Correction for multiple testing was performed using the Bonferroni method (Padj). Analyses were performed using SPSS (IBM Corp. Released in 2016. IBM SPSS Statistics for Windows, Version 25.0. Armonk, NY).

## 3. Results

### 3.1. Study Population

Between February 2014 and October 2015, 114 MS patients and 30 HC were recruited (Figure 1). Age, sex, comorbidities, and (non-immunomodulating) medication use did not significantly differ between these groups. Demographics, clinical variables, self-reported relapses, systemic corticoids, and treatment escalation during follow-up are summarized in Table 2 and Table 3.

Comparative analysis confirmed that the MS and HC groups were comparable in terms of age and sex distribution. The comparison between MS subgroups at baseline, however, revealed a significantly higher age in PPMS than in RRMS_U (*p* = 0.003) and in RRMS_I (*p* = 0.004), as well as higher NFL levels in PPMS than in BMS (*p* = 0.004) and RRMS_I (*p* = 0.002) in post-hoc analysis. EDSS was lowest in BMS (*p* < 0.05) and highest in PPMS (*p* < 0.001) compared to other MS subgroups, while disease duration was longer in BMS than in RRMS_U (*p* = 0.001) and PPMS (*p* = 0.039). ARMSS was significantly higher in PPMS than in BMS (*p* < 0.001) and RRMS_U (*p* = 0.017).

### 3.2. Heart Rate Variability and Serum Samples

Three ECG recordings from HC at baseline were excluded because they contained >20% ectopic beats. All ECG samples at month three were eligible (*n* = 97, NMSC samples only). Two blood samples were lost during processing. Of the remaining 132 serum samples (97.8%), 35 samples (26.5%) had CRP levels under the detection limit (<0.50 mg/L). These were given the value of 0.0 mg/L for statistical purposes. All had valid NFL values (Table 2).

### 3.3. Cross-Sectional Analysis

SDNN and RMSSD did not significantly differ between MS and HC (F = 1.029, Padj >0.1 and F = 0.229, Padj > 0.1, respectively), nor between MS subgroups (F(3, 105) = 2.374, Padj > 0.1 and F(3, 105) = 2.319, Padj = 0.08, respectively). At visual inspection of the data, however, we did remark a non-significantly higher SDNN and RMSSD at baseline in RRMS_U and BMS than in RRMS_I and PPMS, which was mirrored in the month three SDNN results. SDNN and RMSDD were strongly correlated at baseline (r = 0.884, Padj < 0.001). They were both inversely correlated with age (r = −0.278, Padj = 0.018 and r = −0.319, Padj < 0.001, respectively), EDSS (r = −0.254, Padj = 0.042 and r = −0.240, Padj = 0.01, respectively), and serum NFL (r = −0.251, Padj = 0.048 and r = −0.177, Padj = 0.04, respectively) in the whole MS population. A partial correlation test revealed that the correlation between SDNN/RMSSD and serum NFL was explained by a collinear effect with age (partial r = −1.31, *p* > 0.1 and partial r = −0.077, *p* > 0.1, respectively). There were no significant correlations between HRV indices and sex, ARMSS, and serum CRP (Padj > 0.1).

### 3.4. Longitudinal Analysis

In MS patients, SDNN at baseline and three months correlated strongly over time at the group level (r = 0.695, *p* <0.001), but had limited intra-individual reproducibility (Z = −2.343, *p* = 0.019). Findings were similar for RMSSD at baseline and month 3 (r = 0.760, *p* <0.001 and Z = −2.211, *p* = 0.027). The two SDNN outliers in Figure 2 reported a relapse within three (RRMS_U) or twelve months (BMS) after baseline. These outliers are the same as for RMSSD. Exclusion of these outliers from analysis did not change results.

In MS patients, SDNN at baseline was significantly related to relapse occurrence at month 3 in univariate (OR = 1.048, 95% CI (1.013–1.085), *p* = 0.007) and multivariate analysis (OR = 1.053, 95% CI (1.013–1.095), *p* = 0.009). The same was true for relapse occurrence at month 12 in univariate (OR = 1.031, 95% CI (1.009–1.054), *p* = 0.007) and multivariate analysis (OR = 1.034, 95% CI (1.009–1.059), *p* = 0.008). This is confirmed by an Independent samples t test wherein SDNN at baseline was compared between MS patient that either had of did not have a relapse at month 12 (Independent samples t test, F = 0.190, *p* = 0.005; Appendix A).

The logistic regression analysis was repeated with RMSSD at baseline and relapse occurrence during the first 3 months (univariate OR = 1.051, 95% CI (1.014–1.089), *p* = 0.007; and multivariate OR = 1.065, 95% CI (1.016–1.117), *p* = 0.009); and the total 12 months (univariate OR = 1.023, 95% CI (1.000–1.047), *p* = 0.047); and multivariate OR = 1.028, 95% CI (0.999–1.057), *p* = 0.061).

Age, sex, MS subgroups allocation and age-corrected EDSS and NFL did not significantly contribute to either multivariate model (*p* > 0.1).

The ROC curve for SDNN at baseline and the occurrence of relapses during twelve months is shown in Figure 3 (AUC = 0.733, *p* = 0.002).

## 4. Discussion

The present study did not find significant differences in SDNN or RMSSD between MS and HC, nor between MS subgroups. Some studies indicate lower SDNN and RMSSD in MS (progressive and RRMS [10] or RRMS alone [12,13]) than in HC, and others found higher SDNN in RRMS than in progressive MS [10,15]. Differences in samples, methodology, and immunomodulatory treatment make it difficult to compare these results. Additionally, proper age-matching is needed to differentiate between disease-related pathophysiology and ageing [16,17]. The observed correlation between SDNN/RMSSD and NFL at baseline was explained by a collinear effect with age. Repeated SDNN measures showed a strong correlation at the group level, but were not individually reproducible, illustrating the variable nature of HRV.

The positive relationship between baseline SDNN and RMSSD with relapses was surprising since it contrasts with the hypothesized health-protective effects of the vagal nerve in chronic diseases [22]. This has not been described previously in MS. Furthermore, this relationship appears to be largely driven by their association with relapses in the first three months, since the significant longitudinal effect of RMSSD is restricted to relapse occurrence during the first three months. Following the theory that “T-cell immunity to self maintains the self” [33], we speculate that the vagus nerve may have increased its anti-inflammatory activity [34] in response to raised sub-clinical CNS inflammation in relapsing MS patients, which is supported by the observed anti-inflammatory effect of vagal stimulation in mice studies on injury-induced inflammation [35]. Relapse occurrence could thus reflect a failing or insufficient immune response. The lack of a correlation between SDNN/RMSSD and CRP at baseline suggests no relationship with peripheral inflammation in this population. Finally, the positive relationship between HRV and relapses observed here contradicts our observation that HRV was inversely related to NFL.

While the relation between SDNN/RMSSD and relapses currently remains unexplained, we propose to further investigate HRV as a marker of CNS inflammation in MS, in view of the anti-inflammatory role of the vagus nerve [34].

We included patients with distinct MS disease courses [27] to maximize differences in presumed inflammatory disease activity. The treated group was limited to interferon treatment. BMS was included as a separate group, which has not been done in previous HRV studies in MS. A prospective one-year follow-up allowed to examine the contribution of HRV to short-term disease evolution. Serum biomarkers were collected to search for an underlying mechanism linking parasympathetic activity and MS outcomes. Age distributions between MS and HC groups were matched and age was added to the logistic regression model.

The study had a few limitations. Patient classification in MS has limitations. Strict inclusion criteria for subgroups did not prevent that six patients started treatment shortly after inclusion. Most probably, this decision was taken before recruitment. The HC group was small compared to the whole MS sample. We used self-reported relapses, which were not necessarily confirmed by a physician. Nonetheless, they have been used as a patient-reported outcome in other studies [36,37], with a tendency towards underreporting [38]. We could not perform a cox regression analysis because the exact timing of relapses was unknown. We did not control for diurnal variations in HRV. SDNN levels in our HC group were lower and less variable than in Mahovic’s study (mean 30 ± 22.05 ms versus 135 ± 24 ms [12]), which could be explained by the higher age and lower rate of ectopic beats in our HC population [39]. While the intergroup analysis did not show any significant results, our study was underpowered to detect significant SDNN/RMSSD differences between these MS subgroups. Therefore, any MS intergroup difference in SDNN/RMSSD cannot be ruled out by these results alone. Lastly, not all MS participants had MRI scans during the study and those that had, did not obtain them within a predetermined time window. Therefore, we were not able to examine the relationship between HRV and MRI results.

Nevertheless, this study included a relatively large MS sample and a prospective design, which considered multiple confounders in the analysis. We recommend further exploration of the evolution of HRV in a larger sample of RRMS patients before, during and after a physician-confirmed relapse, and include longer prospective follow-up in RRMS and PPMS patients, including repeated measurements of serum and CNS biomarkers for inflammation. This would improve our understanding of how HRV interacts with acute and chronic CNS inflammation and possibly relates to the disease course. Whether HRV and its relation to inflammation may have a prognostic effect on disease severity, as in cancer [40,41], could be examined in MS as well.

## 5. Conclusions

HRV did not significantly differ between MS patients and healthy controls, nor between predefined MS subgroups. However, a higher baseline SDNN and RMSSD was associated with relapse occurrence during follow-up, most pronounced during the first three months. Our findings suggest a link between HRV and ongoing disease activity in MS. We propose to investigate HRV and its relationship with physician-confirmed relapses in long-term prospective studies to improve our understanding of the link between vagal tone, neuro-inflammation, and the course of MS.

## Figures and Tables

**Figure 1 jcm-09-00003-f001:**
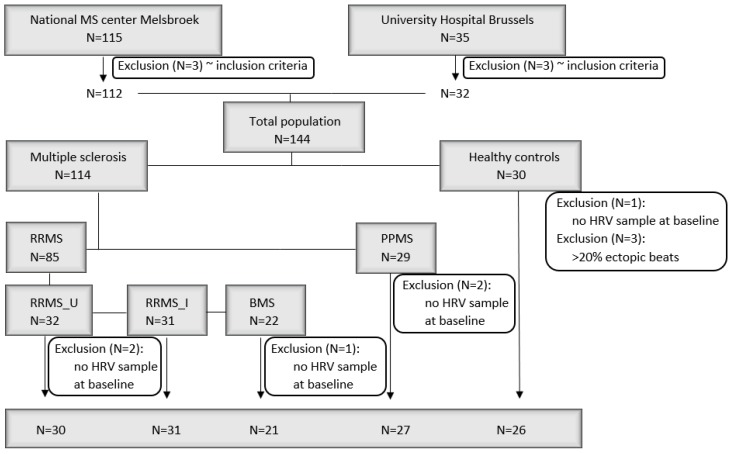
Overview of the study population. BMS: benign multiple sclerosis; HRV: heart rate variability; PPMS: primary-progressive multiple sclerosis; RRMS_I: interferon-treated relapsing-remitting multiple sclerosis; RRMS_U: untreated RRMS.

**Figure 2 jcm-09-00003-f002:**
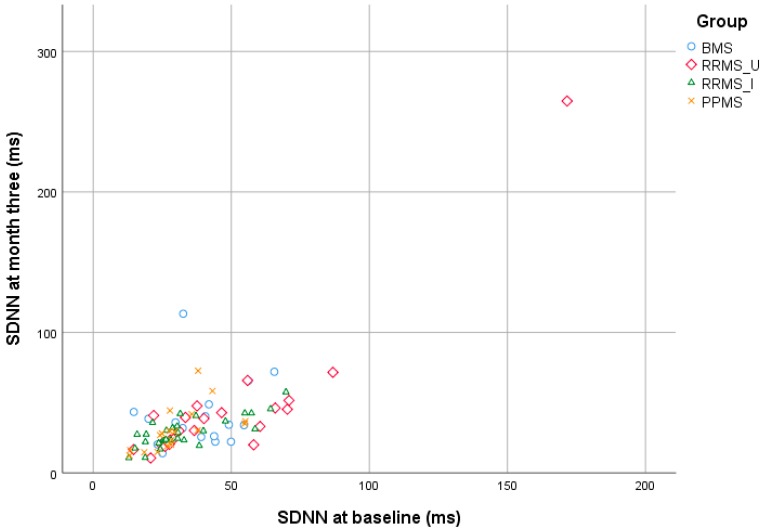
Scatterplot comparing SDNN values at baseline and after three months. The two datapoints >100 ms on the Y-axis are considered outliers and are explained in the text. BMS: benign multiple sclerosis; HRV: heart rate variability; PPMS: primary-progressive multiple sclerosis; RRMS_I: interferon-treated relapsing-remitting multiple sclerosis; RRMS_U: untreated RRMS; SDNN: standard deviation of each normal-to-normal inter-beat interval.

**Figure 3 jcm-09-00003-f003:**
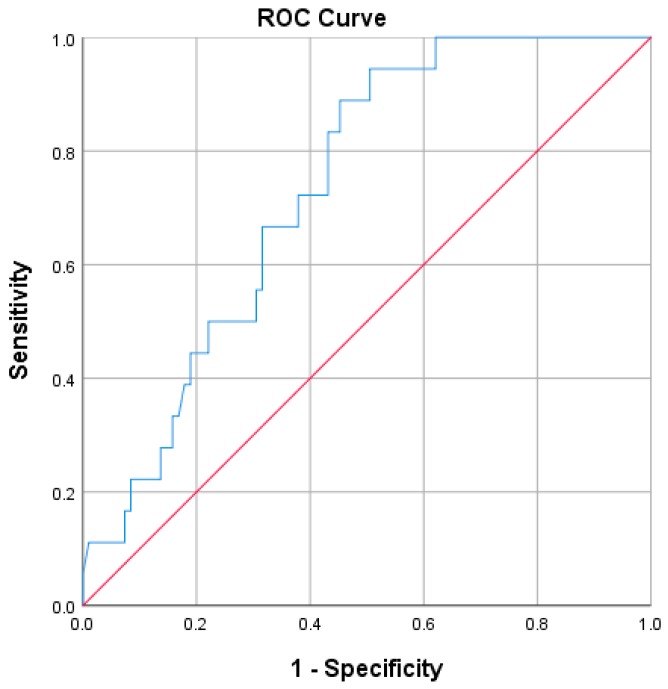
ROC curve illustrating sensitivity and specificity of SDNN at baseline on the occurrence of self-reported relapses (AUC = 0.733, *p* = 0.002). Diagonal segments are produced by ties. AUC: area under the curve; ROC: receiver operating characteristic; MS: multiple sclerosis; SDNN: standard deviation of each normal-to-normal inter-beat interval.

**Table 1 jcm-09-00003-t001:** Inclusion- and exclusion criteria. EDSS: expanded disease status scale.

**Inclusion Criteria for Patients**	
Untreated active relapsing-remitting multiple sclerosis (RRMS_U)	≥1 clinical relapse 2 years prior to screening or ≥1 active contrast-enhancing lesion on brain MRI in the last year EDSS < 7.0
Interferon-treated relapsing-remitting multiple sclerosis (RRMS_I)	≥3 months of stable treatment with interferon-bèta EDSS < 7.0
Untreated benign multiple sclerosis (BMS)	EDSS ≤ 3.0 at least 15 years after first symptoms [28]
Non-active primary-progressive multiple sclerosis (PPMS)	No clinical relapse within 2 years prior to screening EDSS < 7.0
**Exclusion Criteria for Patients and Healthy Controls**	
Secondary progressive multiple sclerosis or other diseases of the central nervous system Immunomodulatory drugs other than interferon-bèta at screening Treatment with glatiramer acetate ≤3 months, fingolimod or natalizumab ≤6 months, or systemic corticosteroids ≤2 months prior to screening	

**Table 2 jcm-09-00003-t002:** Demographics, clinical and biochemical data at baseline. Raw (P) and adjusted *p*-values (Padj) are reported (Bonferroni). ARMSS: age-related multiple sclerosis severity; EDSS: expanded disease status scale; CRP: C-reactive protein; Iqr: interquartile range; NA: not applicable; NFL: neurofilament light; SD: standard deviation; SDNN: standard deviation of the normal-to-normal inter-beat interval.

	HC (*n* = 26)	MS (*n* = 109)	Padj	RRMS_U (*n* = 30)	RRMS_I (*n* = 31)	BMS (*n* = 21)	PPMS (*n* = 27)	P	Padj
Mean age (SD) in years	49.6 (9.3)	46.7 (9.3)	>0.1	43.3 (10.9)	44.2 (7.7)	47.3 (6.4)	52.7 (8.6)	0.001	0.009
Female (%)	65.4	64.2	>0.1	70.0	77.4	66.7	40.7	0.025	>0.1
Median EDSS (iqr)	NA	3.0 (3.0)	NA	2.0 (2.0)	3.0 (2.0)	2.0 (1.0)	5.5 (2.5)	<0.001	<0.001
Median ARMSS (iqr)	NA	4.7 (4.1)	NA	4.9 (4.0)	4.7 (3.8)	2.3 (1.6)	6.8 (3.3)	<0.001	<0.001
Median disease duration in years (iqr)	NA	13.0 (13.0)	NA	6.0 (12.0)	11.0 (11.0)	17.0 (6.0)	8.0 (12.0)	0.001	0.009
Median SDNN at baseline in ms (iqr)	30.0 (22.1)	37.7 (25.6)	>0.1	37.1 (39.7)	28.7 (14.7)	40.6 (22.1)	27.7 (20.3)	0.076	>0.1
Median RMSSD at baseline in ms (iqr)	18.0 (20.2)	20.8 (19.1)	>0.1	28.2 (34.0)	18.3 (13.2)	21.2 (17.6)	16.9 (15.7)	>0.1	>0.1
Median CRP in mg/L (iqr)	0.9 (2.3)	1.0 (2.9)	>0.1	1.1 (3.2)	0.8 (2.6)	0.8 (4.2)	1.0 (2.7)	>0.1	>0.1
Median NFL in ng/L (iqr)	10.1 (4.1)	12.9 (6.9)	>0.1	11.8 (6.0)	10.5 (7.7)	10.1 (6.0)	15.8 (9.3)	0.001	0.009

**Table 3 jcm-09-00003-t003:** Prospective follow-up data. Only results for those with repeated HRV registration at month three are shown. Adjusted *p*-values (Padj) are reported (Bonferroni). Frequencies are not cumulative. IMD: immunomodulatory drug; Iqr: interquartile range; NA: not applicable; SDNN: standard deviation of the normal-to-normal inter-beat interval.

	HC (*n* = 10)	MS (*n* = 87)	Padj	RRMS_U (*n* = 22)	RRMS_I (*n* = 30)	BMS (*n* = 17)	PPMS (*n* = 18)	Padj
**HRV variables at month three**								
Median SDNN month three (ms)	33.1 (25.1–36.8)	29.9 (29.4–42.1)	>0.1	35.9 (22.7–68.4)	27.5 (24.9–32.8)	34.3 (28.1–53.0)	28.7 (22.9–38.7)	>0.1
Median RMSSD month three in ms (iqr)	18.3 (7.8)	19.3 (19.7)	>0.1	25.3 (25.1)	20.3 (18.8)	16.1 (27.5)	17.1 (16.2)	0.08
**Clinical variables at baseline–month three**								
Self-reported relapses	NA	7	NA	4	2	1	0	NA
Systemic corticoid use	NA	1	NA	1	0	0	0	NA
IMD escalation	NA	7	NA	3	1	3	0	NA
**Clinical variables at month three–month twelve**								
Self-reported relapses	NA	9	NA	2	4	3	0	NA
Systemic corticoid use	NA	0	NA	0	0	0	0	NA
IMD escalation	NA	8	NA	5	3	0	0	NA

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
