# Peer review of "Relation between Heart Rate Variability and Disease Course in Multiple Sclerosis"

_jcm, 2019, doi:10.3390/jcm9010003_

Round 1

Reviewer 1 Report

No further comments. I am still worry if the result "No significant HRV differences were found b/w MS and HC, or b/w subgoups" can be a significant output of a research paper.

Author Response

We thank the reviewer. While we understand his/her worry, our exploratory study was underpowered to detect minor intergroup differences. The stratification of patients suffering from a complex spectrum disease like MS was not an easy task and was not very helpful in the search for intergroup differences. Our additional finding of relapse occurrence being associated with increased HRV across all MS subgroups needs to be replicated and validated, as this is a promising finding.

Reviewer 2 Report

The reviewer would like to thank the authors for the attention they paid to the suggestions and concerns. The manuscript  improved in its actual form.

One comment remains. In table 2, the rationale for presenting raw p values sometimes and adjusted p values sometimes does not seem clear. The authors may want to keep both columns but may need to fill both of them, or explain the rationale for alternating in their presentation.

The same applies to section 3.3. It is assumable that all presented p values are adjusted. If that is the case, please replace the displayed p with padj when applicable.

Author Response

We thank the reviewer for the positive review. In the previous peer review round, one of the reviewer requested to add raw p values for demographic characteristics (since it is important to check if any intergroup differences in age, sex, disease duration and EDSS would account for any significant differences on the variables of interest) and adjusted p values for CRP, NFL and HRV parameters. As requested, we now filled in both columns of Table 2 and all adjusted p values in section 3.3 are now displayed with Padj.

This manuscript is a resubmission of an earlier submission. The following is a list of the peer review reports and author responses from that submission.

Round 1

Reviewer 1 Report

I've reviewed the paper

1) The objective is unclear. It is unclear why the analysis of HRV in MS subjects has been applied. Results do not show any significant difference. Moreover methods for HRV analysis are very limited  in respect to literature. No other features than SDNN are used.

2) Why the author did not use other indices in time domain (well known in literature)? Why not the analysis in the frequency domain? 

3) Which algorithm has been adopted to extract HRV from ECG signals? Data about validation?

4) Results did not show any difference between MS and Healthy subject and inside MS subgroups. This means that the analysis does not provide any information.

5) A deeper analysis is required to investigate if changes in HRV could be found in MS patient groups

6) Information provided in tables is presented in a quite connfsed way. Moreover Figure 2 does not help understanding nor describing results.

In summary, both the quality and the scientific content are weak and do not present novelty aspects. The used methods are well known and the application is insufficient. Application to MS subject has been already studied.

Reviewer 2 Report

It is interesting study that compares the heart rate variable measure (i.e., standard deviation of normal-to-normal inter-beat intervals (SDNN) in a relatively large multiple sclerosis (MS) sample (n=109) with different  phenotypes (treated and treatment-naïve relapsing remitting (RRMS), primary progressive (PPMS), and benign course) and (n=26) age- and sex-matched healthy controls. The study also performed group comparisons for SDNN at 3 months follow-up, and assessed in patients, the contribution of SDNN to the risk of having a relapse at one year (logistic regression analysis). Baseline biomarkers (CRP and NFL) were additionnaly obtained in patients. No significant group differences were observed regarding SDNN. However, baseline SDNN predicted the relapse rate at one year. The study adds some value to the available literature on this matter (i.e., subjective relapse rate, diurnal variation in SDNN,). Some of the limitations were already accounted for in the discussion. Other issues remain to be addressed.

Abstract:

It would be helpful to specify that, in the whole MS cohort, SDNN correlated with age (r=-0.278, P=0.018). The same applies to the following sentence : “At the group level, SDNN remained stable over three months (r=0.695, P<0.001)”. Regarding the stability of SDNN, the author used group comparison (Wilcoxon) and correlation analysis to study the variables between baseline and at 3 months. Correlation analysis might not be the optimal statistical test to assess the ‘stability’ of SDNN at 3 months. The correlation analysis rather implies that, regardless of the potential variations, baseline measures were strongly (r>0.60) and significantly (p<0.001) correlated with those obtained at 3 months. Furthermore, Wilcoxon test seems to yield significant results (Z=-2.343, P=0.019), which implies that the results significantly difference at 3 months compared to baseline. Therefore, the statement on SDNN stability might better be avoided here, or rather paraphrased.

Introduction:

It is useful to provide a citation supporting the statement in the first two sentences.

Methods:

Although it is interesting to stratify patients in 4 clinical subgroups, the rationale for the inclusion/exclusion criteria does not seem clear. In other words, it is important to clarify why patients with secondary progressive were excluded. Including those patients, would have allowed to compare the SDNN between patients with early stages of the disease (RRMS) where the hallmark is inflammation, and late stages of the disease (SPMS) where the hallmark is neurodegeneration. The same applies to the decision of only including patients receiving interferons. It is assumable that the authors did so to obtain a homogeneous population in terms of treatments.

In p. 2, it is stated that ‘HC were recruited to match the age- and gender distributions of the MS samples’. It is helpful to clarify how the matching was done (HC vs. the whole MS cohort; or HC vs. each MS subgroup).

The obtained measures in MS patients and HC are sometimes difficult to follow. It is important to clarify in this section that measures CRP and NFL were only obtained in MS patients at baseline. In addition, SDNN were obtained in MS patients and HC at baseline and 3 months later. Correlation analysis was only performed in patients.   

For the logistic regression, it would be interesting to clarify how the collinear variables EDSS and NFL were added as an expression of age.

It is crucial to account for multiple comparison to reduce the risk of type 1 error as the authors did (Bonferroni adjustment). However, when reporting sociodemographic and clinical characteristics, raw p values rather than adjusted p values are usually presented, since it is important to check if any intergroup differences in age, sex, disease duration and EDSS would account for any significant differences on the variables of interest (e.g., SDNN). Therefore, it is advisable to display raw p values for age, sex, EDSS, ARMSS, and report adjusted p values for SDNN, CRP and NFL.

The relapse was reported at 3 months and at 12 months. However, the binary logistic regression model only used relapse at 12 months as a dependent variable. This needs to be clarified. The authors might not have studied the relationship at 3 months because there were no differences between baseline and 3 months SDNN values.

When displaying the results (tables) it is important to state if the results are displayed as mean ± SD (range) or median (interquartile range or range).

In section 3. The following MDPI instruction needs to be deleted: ‘This section may be divided by subheadings. It should provide a concise and precise description of the experimental results, their interpretation as well as the experimental conclusions that can be drawn.’

From the flow chart, it seems that an important part of the data was missing at baseline. It is important to explain the reason if possible and state if/how such missing data constitute any potential inclusion bias.

Results:

Table 2 shows significant differences in age, disease duration, EDSS, ARMSS, and NFL. A relevant section would be added to state the post-hoc details of the subgroups MS comparison.

Section 3.2.

For the following sentence, it is important to state that this is about the HC as the flow chart implies : ‘Three ECG recordings at baseline were excluded because they contained >20% ectopic beats.’ This section should account for the lacking samples as shown in the flow chart. It is unclear how ECG samples in MS patients decreased from 135 samples (n=109 in MS patients and n=26 HC) at baseline to n=97 at 3 months. This warrants an explanation. It might be better to split n=97 in table 2 at 3 months to show the sample size for patients and HC, and the sample size of each MS subgroup.

Section 3.3

ANOVA equations could be clarified by including the degrees of freedom. In table 2, the ANOVA p value for MS subgroup comparison was > 0.1 and in the paragraph it is written as 0.07.

Section 3.3. Longitudinal Analysis should be corrected to 3.4. In this section, it is important to specify that the analysis was conducted only in patients, as the one can understand from the paragraph and the figure. If that is not the case, it is important to clarify.

Discussion and limitations:

In the discussion l. 177, the following statement might be avoided or rather supported by an argument since the authors did not statistically account for age when studying the relationship between SDNN and NFL levels (e.g., partial correlation analysis): ‘The perceived correlation between SDNN and NFL at baseline was explained by a collinear effect with age’.

Using neuroimaging would have allowed to unravel the relationship between cerebral/spinal lesions and the studied parameters representing ANS dysfunction; especially that in one of the cited works, ANS variables were correlated with brainstem pathologies.

References:

Please correct ref [2] [5] by adding the journal name. References [23] and [24] do not seem to constitute publications but rather a reference to the used material, and therefore could be only cited in the text without referring to it in the reference list.

Reviewer 3 Report

The paper presents the relationship between HRV and multiple sclerosis activity. 144 subject data sets were incorporated for this study and quite good methods and analysis were conducted. However, even with the nice analysis, the results and the conclusion might not be interesting and attractive to the readers. The results is that HRV is not related to the MS activity. In there, the reviewer assume that the hypothesis is not correctly defined or the motivation of the study is not appropriately considered. Where, for example, HRV can be varying in terms of normal person’s routine life while exercise or physiological condition changes, then, how the HRV can be a measure of MS disease course? That is also related to the contribution and significance of the study. Please validate and elaborate this concerns.